

# Dynamic proteomic changes in soft wheat seeds during accelerated ageing

Yangyong Lv, Pingping Tian, Shuaibing Zhang, Jinshui Wang and Yuansen Hu

College of Biological Engineering, Henan University of Technology, Zhengzhou, China

## ABSTRACT

Previous research demonstrated that soft wheat cultivars have better post-harvest storage tolerance than harder cultivars during accelerated ageing. To better understand this phenomenon, a tandem mass tag-based quantitative proteomic analysis of soft wheat seeds was performed at different storage times during accelerated ageing (germination ratios of 97%, 45%, 28%, and 6%). A total of 1,010 proteins were differentially regulated, of which 519 and 491 were up- and downregulated, respectively. Most of the differentially expressed proteins were predicted to be involved in nutrient reservoir, enzyme activity and regulation, energy and metabolism, and response to stimulus functions, consistent with processes occurring in hard wheat during artificial ageing. Notably, defense-associated proteins including wheatwin-2, pathogenesis-related proteins protecting against fungal invasion, and glutathione S-transferase and glutathione synthetase participating in reactive oxygen species (ROS) detoxification, were upregulated compared to levels in hard wheat during accelerated ageing. These upregulated proteins might be responsible for the superior post-harvest storage-tolerance of soft wheat cultivars during accelerated ageing compared with hard wheat. Although accelerated ageing could not fully mimic natural ageing, our findings provided novel dynamic proteomic insight into soft wheat seeds during seed deterioration.

## INTRODUCTION

Wheat (*Triticum aestivum L.*), one of the most important crops in the world, is a staple food used for making flour, noodles, alcoholic beverages, biofuel and a variety of other products (*Lim, 2013*). The post-harvest storage of wheat seeds is commonly carried out in horizontal warehouses for 3 to 5 years or longer in several northern provinces in China. As storage time is prolonged, seeds can deteriorate via incremental decomposition of stored substances, and impaired metabolism and energy supply, ultimately leading to commercial losses. Two of the most significant environmental factors affecting this process are temperature and relative humidity (*McDonald, 1999*). Extensive studies have been conducted to investigate physiological and biochemical changes during seed ageing. These researches indicate that accumulated free radicals attack membrane lipids, triggering degenerative events including changes in lipid content, nutrients reserves, lipoxygen activity and nucleic acids (*Hu et al., 2012*; *McDonald, 1999*; *Rehman & Shah, 1999*).

To study seed deterioration, accelerated ageing of seeds at high temperature and relative humidity for short durations was performed to evaluate storage potential. This

Corresponding author
Yuansen Hu,
huyuansen@haut.edu.cn,
hys308@126.com

method has been extensively used by researchers to investigate natural ageing (*Galleschi et al., 2002*; *Calucci et al., 2004*; *Xin et al., 2011*). Several researchers concluded that physiological changes in seeds subjected to accelerated ageing were the same as those occurring during natural ageing, only the rate differs, whereas others believe that artificial ageing treatments cannot replicate the actual seed status during storage, and question whether physiological changes during accelerated ageing reflect events during natural ageing (*McDonald, 1999*; *Rajjou et al., 2008*; *Walters, 1998*). In wheat, kernel texture and biochemical composition vary with the degree of hardness (*Lesage et al., 2011*; *Pasha, Anjum & Morris, 2010*). Previous researches showed that seed ageing causes the physiological and biochemical variation including differences in hardness during storage, and these changed characteristics have been extensively studied, specifically changes in antioxidants, free radicals, storage proteins, and respiration (*Galleschi et al., 2002*; *Calucci et al., 2004*; *Keskin, Yalçın & Özkaya, 2018*; *McDonald, 1999*; *Zhang et al., 2017*). Previous studies also indicate that the seed ageing effects the proteome of dry seeds (*Rajjou et al., 2008*; *Xin et al., 2011*). Additionally, proteomic changes occur in medium-hard wheat seeds of the 'Aikang58' cultivar during deterioration, as evidenced by 162 differentially expressed proteins (DEPs), confirming proteome variation in dry seeds during artificial ageing (*Lv et al., 2016*). Most previous studies have focused on quality, physiological and biochemical changes during storage, whereas dynamic quantitative proteomic changes in wheat seeds, especially those in soft wheat seeds that take place during accelerated ageing remain to be elucidated.

Proteomic approaches including two-dimensional (2-D) electrophoresis and 2-D differential gel electrophoresis have been extensively used to probe seed physiology, but these gel-based methods suffer from low throughput, poor reproducibility and limited ability to detect low abundance and hydrophobic proteins (*Ge et al., 2013*; *Mak et al., 2009*). These disadvantages can be overcome by non-gel and mass spectrometry-based quantitative proteomic approaches using isobaric tagging reagents, such as tandem mass tags (TMT) and isobaric tags for relative and absolute quantification (iTRAQ) (*Karp et al., 2010*; *Van Ulsen et al., 2009*). In this work, we conducted the first tandem mass tag (TMT)-based dynamic quantitative proteomic analysis of the seeds of the elite Chinese soft wheat cultivar 'Yangmai 15' during artificial ageing. The identified DEPs were associated with a range of different functions and provided new insight into the comprehensive understanding of deterioration in soft wheat seeds.

## MATERIALS AND METHODS

### Wheat seeds and artificial ageing treatment

The 'Yangmai 15' soft cultivars used in this study were purchased from the Henan Academy of Agricultural Science. The original germination rate (Gr) was 97.0% (designated as YM97), and the average seed moisture content was 12.19%. A 400 g sample of seeds with a similar size and weight was stored in a 5 L plastic bottle at 45 ± 1 °C and 50% relative humidity in a constant temperature- and humidity-controlled cabinet (Binder KMF720; Binder, Tuttingen, Germany) according to our previous research (*Lv et al.,*

*2016*). Measurement of $CO_2$ from wheat respiration and calculation of Gr were performed as previously described (*Dong et al., 2015*; *Zhang et al., 2014*). Gr and the $CO_2$ concentration were determined regularly and seed samples with three lots of biological replicates were collected for further proteomic analysis when Gr was 97%, 45%, 28% and 6% (designated as YM97, YM45, YM28 and YM6, respectively).

## Protein extraction, SDS-PAGE analysis, trypsin digestion, TMT labelling and high-pressure liquid chromatography (HPLC) fractionation

Protein extraction, trypsin digestion and HPLC fractionation were performed as reported previously (*Lv et al., 2016*). Extracted proteins were boiled in loading buffer (100 mM Tris-HCl buffer (pH 8.0) containing 2% SDS, 5% glycerol, and 1% β-mercaptoethanol) for 5 min. They were centrifuged at $12,000 \times$ g for 5 min, and supernatants were subjected to 12% SDS-PAGE in a vertical slab gel apparatus (LIUYI, Beijing China). After trypsin digestion, peptides were desalted using a Strata X C18 SPE column (Phenomenex, Torrance, CA, USA) and vacuum-dried. Peptides were reconstituted in 0.5 M triethylammonium bicarbonate and processed according to the manufacturer's protocol for the TMT kit (Thermo Fisher Scientific, Waltham, MA, USA). Briefly, one unit of TMT reagent was thawed and reconstituted in acetonitrile, and peptides from YM97, YM45, YM28, and YM6 (labelled with 126, 127, 130, and 131, respectively) were then incubated with TMT reagent for 2 h at room temperature and pooled, desalted and dried by vacuum centrifugation. Labeled samples combined for further analysis.

## Liquid chromatography tandem mass spectrometry (LC-MS/MS) analysis and database searches

LC-MS/MS and database searches were performed as described previously (*Lv et al., 2016*). Briefly, peptides were dissolved in 0.1% formic acid (solvent A) and separated using a reversed-phase analytical column (Acclaim PepMap RSLC; Thermo Fisher Scientific, Waltham, MA, USA). The gradient comprised an increase from 6% to 23% solvent B (0.1% formic acid in 98% acetonitrile) over 26 min, 23% to 35% in 8 min, an increase to 80% in 3 min, and holding at 80% for 3 min, all at a constant flow rate of 400 nL/min on an EASY-nLC 1000 UPLC system. The resulting peptides were analyzed and subjected to a nanospray ion source followed by tandem mass spectrometry (MS/MS) in a Q Exactive$^{TM}$ instrument (Thermo Fisher Scientific, Waltham, MA, USA) coupled online to an ultra-performance LC module. Peptides were selected for MS/MS using NCE setting of 28, and fragments were detected in the Orbitrap at a resolution of 17,500. A data-dependent procedure that alternated between one MS scan followed by 20 MS/MS scans with 15.0 s dynamic exclusion. The resulting MS/MS data were processed using the MaxQuant search engine (v.1.5.2.8). Tandem mass spectra were used as queries against the *Uniprot_Triticum_aestivum* database (136,892 sequences) concatenated with the reverse decoy database. The false discovery rate was adjusted to <1%, and the minimum score for peptides was set at >40. Proteins displaying a 1.2-fold change between artificially aged and normal seeds (YM45 vs. YM97, YM28 vs. YM97 and YM6 vs. YM97) were considered as DEPs if $p < 0.05$.
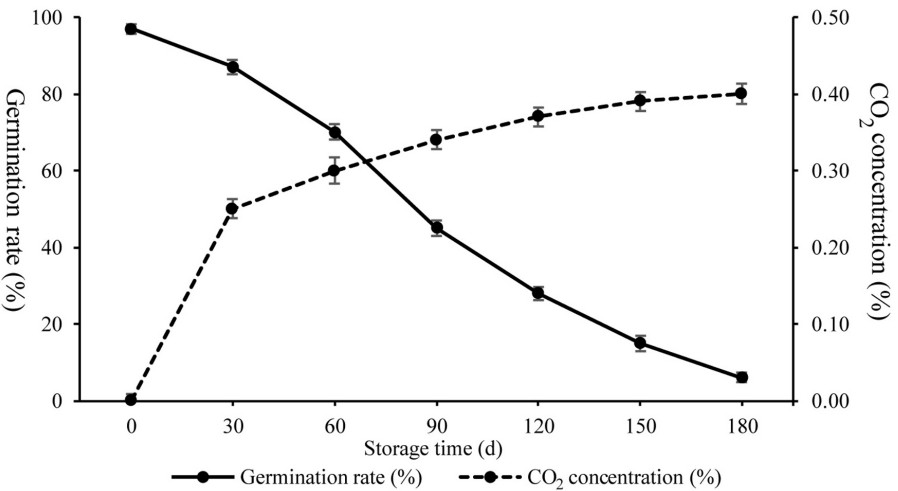

**Figure 1  CO$_2$ concentration and germination rates of wheat seeds during artificial ageing.**

## Protein annotation and functional enrichment analysis

Gene Ontology (GO) annotation and Kyoto Encyclopedia of Genes and Genomes (KEGG) analyses were performed as previously described (*Wu et al., 2006*). Additionally, domain annotation and subcellular localization were performed using InterProScan on InterPro (http://www.ebi.ac.uk/interpro/) and WoLFPSORT systems (PSORT/PSORT version II; http://psort.hgc.jp/), respectively (*Horton et al., 2007*; *Zdobnov & Apweiler, 2001*). Functional-enrichment-based clustering for different protein groups (GO, Domain, Pathway) was performed and cluster membership was visualized using a heat map generated with the "heatmap.2" function in the "gplots" program within the R package as previous reported (*Lv et al., 2016*).

## RESULTS

### Seed respiration and germination rate are altered during accelerated ageing

To characterize physiological changes occurring during artificial ageing, respiration and Gr were investigated (Fig. 1). The Gr of seeds following artificial ageing treatment displayed decreased germinability from 97% to 87% (30 days), 70% (60 days), 45% (90days), 28% (120 days), 15% (150 days), and 6% (180 days), while the amount of CO$_2$ produced increased from 0% to 0.25% (30 days), 0.30% (60 days), 0.34% (90 days), 0.37% (120 days), 0.39% (150 days), and 0.40% (180 days). These results showed that Gr and respiration changed rapidly from 30 d to 90 d indicating drastic seed vigor and metabolic alterations.

### Overview of quantitative proteomics analysis

To characterize the wheat seed proteome, a TMT-based gel-free quantitative proteomics analysis of seeds with three biological replicates was performed during artificial ageing. Sodium dodecyl sulfate-polyacrylamide gel electrophoresis (SDS-PAGE) analysis of proteins from wheat seeds is shown in Fig. S1A. Each lane was loaded with 30 μg of protein,

and proteins of different molecular weight were distributed in each lane, indicating that the protein was not degraded and hence suitable for subsequent analysis. For LC-MS/MS analysis, distribution of peptide length (8–16), mass error (<0.02 Da), and repeatability of relative protein quantitation evaluation (characterized with pair-wise Pearson's correlation coefficients) indicated that MS data accuracy, sample preparation, and reproducibility were acceptable (Figs. S1B–S1D). In this study, 6,271 proteins were identified, among which 4,509 were quantified. Of the quantified proteins, 1,010 displayed significant changes, of which 519 were upregulated (ratio > 1.2; $p < 0.05$) and 491 were downregulated (ratio < 1/1.2; $p < 0.05$) in the artificial aged seeds compared with untreated (control) seeds (Table S1). Of the upregulated proteins, there were 29 in YM45 vs. YM97, 46 in YM28 vs. YM97, and 444 in YM6 vs. YM97; of the downregulated proteins, there were 36 in YM45 vs. YM97, 28 in YM28 vs. YM97, and 427 in YM6 vs. YM97. The above results showed that most of the DEPs distributed in YM6 (Gr was 6%) compared with YM97, indicating that the proteomic profiles were significantly changed during accelerated ageing. The mass spectrometry proteomics data have been deposited to the ProteomeXchange Consortium via the PRIDE (Vizcaíno et al., 2015) partner repository with the dataset identifier hyperlink to PXD009156.

## Protein annotation and functional enrichment analysis of DEPs

To determine the functions of the quantified DEPs, annotation and functional enrichment analyses were performed including subcellular localization, GO, domain, and pathway analysis (Table S1). To better understand the dynamics of DEPs during accelerated ageing, a hierarchical clustering analysis was carried out to obtain dynamic expression patterns (Figs. 2 and 3). The biological process category of GO analysis indicated that upregulated proteins involved in resistance (wheatwin and defensin-like proteins), killing of cells of other organisms, and disruption of cells of other organisms were enriched in YM28. Additionally, upregulated proteins participating in macromolecular metabolic processes were also enriched in YM6, indicating the degradation of storage nutrients (Fig. 2A). Meanwhile, downregulated proteins were mainly involved in regulation of carbon, nitrogen and macromolecular metabolic processes (alpha-amylase/trypsin inhibitor and transcriptional factors) when the Gr was 6%. For the cellular component subclass, upregulated proteins involved in proteasome, endoplasmic reticulum membrane, and extracellular region were enriched in YM6. At the beginning of storage, enriched downregulated proteins were mainly associated with membrane component (YM45), but with an extended artificial ageing time (YM28), enriched downregulated proteins were mainly linked to nucleosome, chromatin and chromosome, and downregulated proteins in YM6 were mainly enriched in respiratory chain, ribosome and ribonucleoprotein complex categories (Fig. 2B). Molecular function analysis indicated that upregulated proteins participated in amylase activity, hydrolase activity, cellulase activity and transporter activity, while down-regulated proteins were involved in peptidase regulatory activity, nutrient reservoir activity, structural constituents of ribosome, enzyme regulator activity and protein dimerization/heterodimerization (Fig. 2C). Protein domain analysis demonstrated that many upregulated proteins contained domains involved in pathogenesis-related
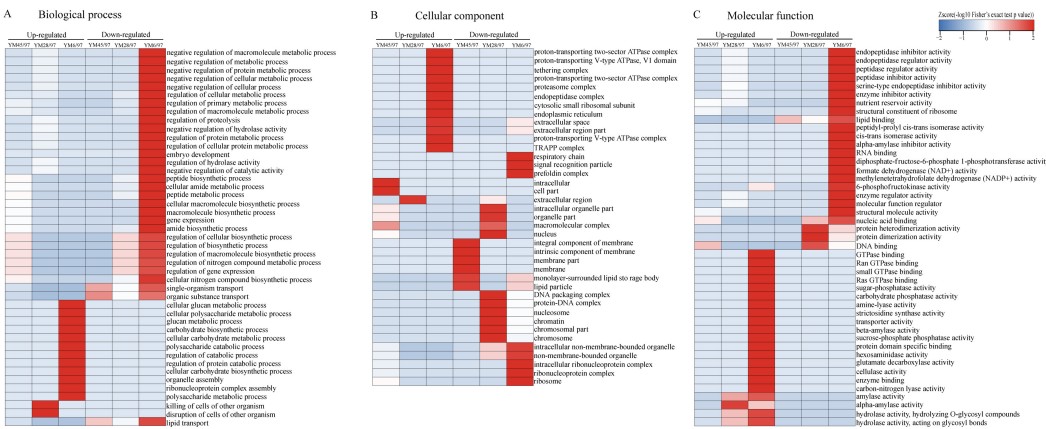

**Figure 2** **Functional-enrichment-based clustering of DEPs (GO analysis).** (A) Biological process, (B) cellular component and (C) molecular function. Each category of DEPs includes both up- and down-regulated proteins. YM45/97 represents proteins displaying significant changes in Gr45% compared with Gr97%; YM28/97 represents proteins displaying significant changes in Gr28% compared with Gr97%; YM6/97 means proteins displaying significant changes in Gr6% compared with Gr97%.

proteins, alpha-amylase and glycosyl hydrolase when Gr was 28%, whereas when Gr was 6%, upregulated proteins were mainly associated with cytochrome c, glycoside hydrolase, and 14-3-3 protein. Additionally, upregulated proteins with a mitochondrial carrier domain were enriched in YM45, proteins with histone fold and histone H2 were abundant in YM28, and proteins with 50S ribosomal protein, proteinase inhibitor, non-specific lipid-transfer protein and bifunctional trypsin/alpha-amylase inhibitor/seed storage domains were enriched in YM6 (Fig. 3A). KEGG analysis indicated that upregulated proteins involved in MAPK signaling pathway, signal transduction and plant pathogen interactions were mainly enriched when Gr was 28% and 6%, and upregulated proteins associated with glycan, amino acids, glutathione metabolism, pentose phosphate pathway and starch and sucrose metabolism were enriched when Gr was 6%. Downregulated proteins related to protein processing in the endoplasmic reticulum were enriched when Gr was 45%, and proteins linked to oxidative phosphorylation and DNA modification were downregulated following extended storage time (Fig. 3B).

## Classification of DEPs during artificial ageing

To better understand the functions of proteins, representative DEPs were classified into six groups involving enzyme activity and regulation, nutrient reservoir activity, electron transport and signal transduction, energy and metabolism, defense/stress responses and chromatin and ribosome (Table 1).

Enzymes involved in starch hydrolytic activity and proteolysis such as alpha/beta-amylase, beta-glucosidase, and proteasome subunit were upregulated, while alpha-amylase/trypsin inhibitor and proteinase inhibitor subtilisin-chymotrypsin were downregulated, which was inconsistent with the increased enzyme activities. Accompanying the increased levels of hydrolytic enzymes was the reduced abundance of storage proteins avenin, gliadin, gamma-hordein and oleosin. Additionally, proteins involved

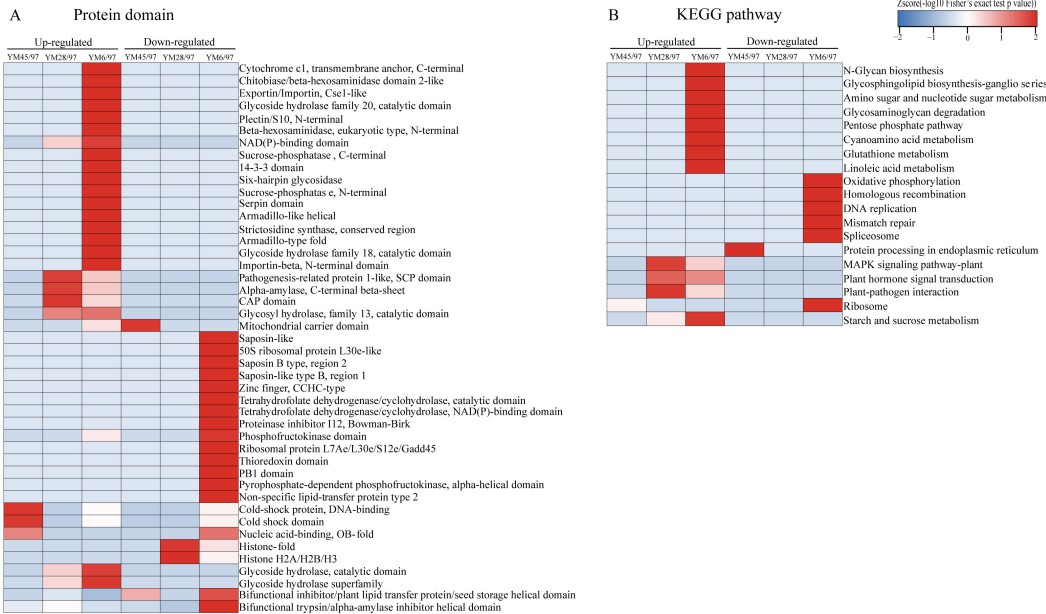

**Figure 3** **Functional-enrichment-based clustering of DEPs.** (A) Protein domain and (B) KEGG pathway. Each category includes both up- and down-regulated DEPs. YM45/97 represents proteins displaying significant changes in Gr45% compared with Gr97%; YM28/97 represents proteins displaying significant changes in Gr28% compared with Gr97%; YM6/97 represents proteins displaying significant changes in Gr6% compared with Gr97%.

in electron transport and signal transduction including cytochrome c oxidase, and calcium-dependent protein kinase and 14-3-3 protein were upregulated. These results also indicated that proteins in energy and metabolism were differently regulated. Two downregulated ATP synthase submits were observed and glyceraldehyde-3-phosphate dehydrogenase that is involved in glycolysis was downregulated, whereas glucose-6-phosphate isomerase, 6-phosphofructokinase, acyl-coenzyme A oxidase, and pyruvate kinase were upregulated. Malate dehydrogenase and succinate dehydrogenase that are involved in the tricarboxylic acid cycle were downregulated, whereas citrate synthase, and isocitrate dehydrogenase were upregulated. Additionally, glucose-6-phosphate 1-dehydrogenase that is involved in the pentose phosphate pathway was upregulated. Among the proteins involved in defense/stress responses, peroxidases and superoxide dismutases, protein disulfide-isomerases, non-specific lipid-transfer proteins, puroindoline-B, and L-ascorbate peroxidase were downregulated, whereas glutathione S-transferase, glutathione synthetase, pathogenesis-related protein, defensin-like protein, wheatwin-2, and heat stress-associated DnaJ were upregulated. During accelerated ageing, proteins participating in chromatin and ribosome organization such as protein H2A, histone H2B, and the 40S and 60S ribosomal proteins were downregulated.

**Table 1** The selected proteins with differential accumulation during seed ageing compared with unaged seeds.

| Protein accession | Protein description | Regulated type | Ratio | Stage | MW [kDa] |
|---|---|---|---|---|---|
| **Enzyme activity and regulation** | | | | | |
| A0A1D5YNU4 | alpha-amylase isozyme 3A | Up | 1.43,1.70 | YM28,YM6 | 52.36 |
| W5FHN7 | alpha-amylase isozyme 3A | Up | 2.04,2.73 | YM28,YM6 | 47.57 |
| A0A1D5YFA7 | Beta-amylase | Up | 1.51 | YM6 | 55.32 |
| A0A1D5XXC9 | Beta-amylase | Up | 1.54 | YM6 | 58.93 |
| A0A1D5XGF4 | Beta-amylase | Up | 1.54 | YM6 | 56.77 |
| W5B347 | beta-glucosidase | Up | 1.22,1.75 | YM28,YM6 | 57.25 |
| W5G0E6 | proteasome subunit alpha type-3 | Up | 1.27 | YM6 | 27.17 |
| W5ENN5 | proteasome subunit beta type-4-like | Up | 1.27 | YM6 | 27.02 |
| A0A1D5XZ64 | proteasome subunit alpha type-5-like | Up | 1.22 | YM6 | 25.88 |
| P01083 | alpha-amylase inhibitor 0.28 | Down | 0.72 | YM6 | 16.80 |
| Q43723 | trypsin/alpha-amylase inhibitor CMX1/CMX3 | Down | 0.81 | YM6 | 13.83 |
| P16159 | Alpha-amylase/trypsin inhibitor CM16 | Down | 0.54 | YM6 | 15.78 |
| A0A1D5UB33 | alpha-amylase/trypsin inhibitor CM1 | Down | 0.72 | YM6 | 21.49 |
| A0A1D5Y0V2 | alpha-amylase/trypsin inhibitor CM3 | Down | 0.57 | YM6 | 17.31 |
| P16851 | Alpha-amylase/trypsin inhibitor CM2 | Down | 0.73 | YM6 | 15.46 |
| P17314 | Alpha-amylase/trypsin inhibitor CM3 | Down | 0.65 | YM6 | 18.22 |
| A0A1D5SLP0 | subtilisin-chymotrypsin inhibitor | Down | 0.49 | YM6 | 9.36 |
| **Nutrient reservoir activity** | | | | | |
| D6QZM5 | avenin-like b6 | Down | 0.83 | YM45 | 32.35 |
| A0A1D6CWE4 | avenin-like b6 | Down | 0.71 | YM6 | 20.58 |
| D2KFH1 | avenin-like a1 | Down | 0.63 | YM6 | 18.92 |
| A0A1D6CWE1 | avenin-like a1 | Down | 0.60 | YM6 | 21.89 |
| P0CZ08 | avenin-like a1 | Down | 0.59 | YM6 | 19.25 |
| M9TGF7 | gamma-gliadin | Down | 0.58 | YM6 | 37.16 |
| M9TK56 | gamma-gliadin | Down | 0.61 | YM6 | 41.20 |
| J7HT09 | alpha-gliadin A-III-like | Down | 0.51 | YM6 | 32.87 |
| P04728 | Alpha/beta-gliadin MM1-like | Down | 0.49 | YM6 | 21.52 |
| A0A1D5S109 | gamma-hordein-3-like | Down | 0.29 | YM6 | 23.67 |
| A0A1D5T3T7 | gamma-hordein-3-like | Down | 0.39 | YM6 | 36.42 |
| W5BE38 | Oleosin | Down | 0.68,0.59 | YM45,YM6 | 16.31 |
| A0A1D5UJY0 | Oleosin | Down | 0.68,058 | YM45,YM6 | 18.22 |
| A0A1D5ZZ07 | Oleosin | Down | 0.83 | YM6 | 16.90 |
| A0A1D6CX50 | Acyl-coenzyme A oxidase | Up | 1.24 | YM6 | 73.97 |
| **Electron transport and signal transduction** | | | | | |
| A0A1D5WWR8 | putative cytochrome c oxidase subunit | Up | 1.25 | YM6 | 8.99 |
| Q6KCK6 | calcium-dependent protein kinase 1 | Up | 1.20 | YM6 | 58.41 |
| G5DFC5 | 14-3-3 protein | Up | 1.24 | YM6 | 29.97 |
| L0GED8 | 14-3-3 protein | Up | 1.32 | YM6 | 29.27 |

**Table 1** (*continued*)

| Protein accession | Protein description | Regulated type | Ratio | Stage | MW [kDa] |
|---|---|---|---|---|---|
| **Energy and metabolism** | | | | | |
| W5BEP1 | ATP synthase subunit d | Down | 0.78 | YM6 | 19.55 |
| W5GLU0 | ATP synthase subunit o | Down | 0.82 | YM6 | 25.57 |
| A0A1D6BD14 | Glyceraldehyde-3-phosphate dehydrogenase | Down | 0.78 | YM6 | 43.73 |
| A0A1D5ST38 | Malate dehydrogenase | Down | 0.81 | YM6 | 35.48 |
| A3KLL4 | Malate dehydrogenase | Down | 0.83 | YM6 | 35.49 |
| A0A1D6BI29 | Succinate dehydrogenase | Down | 0.76 | YM6 | 30.98 |
| W5H1Q1 | Citrate synthase | Up | 1.21 | YM28 | 52.56 |
| A0A1D6RKR9 | Isocitrate dehydrogenase | Up | 1.28 | YM6 | 46.32 |
| A0A1D5UPI1 | Glucose-6-phosphate 1-dehydrogenase | Up | 1.50 | YM6 | 57.82 |
| W5FY62 | Glucose-6-phosphate isomerase | Up | 1.22 | YM6 | 67.06 |
| A0A1D5UP90 | 6-phosphofructokinase | Up | 1.25 | YM6 | 50.73 |
| W5G100 | Pyruvate kinase | Up | 1.49 | YM6 | 57.42 |
| **Defense/stress response** | | | | | |
| A0A1D5S2L6 | peroxidase | Down | 0.71 | YM28 | 37.48 |
| A0A1D5WX80 | peroxidase | Down | 0.80 | YM6 | 35.76 |
| A0A1D5TMJ6 | peroxidase | Down | 0.78,0.81 | YM28,YM6 | 29.77 |
| A0A1D5TAW7 | Superoxide dismutase | Down | 0.78 | YM6 | 21.19 |
| A0A1D5XFA3 | L-ascorbate peroxidase | Down | 0.72 | YM6 | 26.09 |
| Q10464 | Puroindoline-B | Down | 0.75 | YM6 | 16.79 |
| W5D2I6 | Non-specific lipid-transfer protein | Down | 0.80,0.78,0.44 | YM45,YM28,YM6 | 11.23 |
| A0A1D5WSS8 | Non-specific lipid-transfer protein | Down | 0.77,0.43 | YM45, YM6 | 14.81 |
| Q1KMV0 | Non-specific lipid-transfer protein | Down | 0.79,0.77,0.46 | YM45,YM28,YM6 | 11.12 |
| W5D2I6 | Non-specific lipid-transfer protein | Down | 0.80,0.78,0.44 | YM45,M28,YM6 | 11.23 |
| A0A1D5T8E9 | Non-specific lipid-transfer protein | Down | 0.44 | YM6 | 21.11 |
| A0A1D5ZPD0 | Non-specific lipid-transfer protein | Down | 0.45 | YM6 | 12.13 |
| P24296 | Non-specific lipid-transfer protein | Down | 0.45 | YM6 | 11.90 |
| A0A1D5XQK2 | Non-specific lipid-transfer protein | Down | 0.46 | YM6 | 11.97 |
| Q2PCD2 | Non-specific lipid-transfer protein | Down | 0.46 | YM6 | 12.25 |
| P82901 | Non-specific lipid-transfer protein | Down | 0.47 | YM6 | 7.05 |
| A0A1D5UG33 | Non-specific lipid-transfer protein | Down | 0.51 | YM6 | 18.44 |
| A0A1D5Y7A7 | Non-specific lipid-transfer protein | Down | 0.53 | YM6 | 11.86 |
| A0A1D5SG95 | Non-specific lipid-transfer protein | Down | 0.62 | YM6 | 12.61 |
| C3UZE5 | Pathogenesis-related protein 1-like | Up | 1.36 | YM28 | 17.64 |
| A0A1D5YCS8 | pathogenesis-related protein 1-like | Up | 1.85,1.76 | YM28,YM6 | 26.67 |
| P20159 | defensin-like protein 2 | Up | 1.35 | YM28 | 5.15 |
| O64393 | Wheatwin-2 | Up | 1.23 | YM28 | 15.87 |
| P30110 | Glutathione S-transferase 1 | Up | 1.33,1.70 | YM28,YM6 | 25.83 |
| Q8RW01 | Glutathione S-transferase | Up | 1.20 | YM6 | 25.20 |
| W5G485 | Glutathione synthetase | Up | 1.27 | YM6 | 52.9 |

**Table 1** (*continued*)

| Protein accession | Protein description | Regulated type | Ratio | Stage | MW [kDa] |
|---|---|---|---|---|---|
| **Other proteins** | | | | | |
| Q43312 | protein H2A | Down | 0.76,0.56 | YM28,YM6 | 13.93 |
| W5E8H6 | histone H2B | Down | 0.83 | YM28 | 16.31 |
| W5F9T2 | histone H2B | Down | 0.83 | YM28 | 16.49 |
| W5EBF3 | histone H2B | Down | 0.59 | YM6 | 33.13 |
| W5C8N6 | 40S ribosomal protein S6 | Down | 0.56 | YM6 | 28.46 |
| A0A1D5UFM0 | 40S ribosomal protein S12 | Down | 0.72 | YM6 | 15.23 |
| Q7X748 | 60S ribosomal protein L3 | Down | 0.77 | YM6 | 44.62 |
| W5FSM8 | 60S ribosomal protein L17-1 | Down | 0.75 | YM6 | 19.49 |
| A0A1D5X7C5 | Protein disulfide-isomerase | Down | 0.74 | YM6 | 53.25 |
| A0A1D5Y681 | Protein disulfide-isomerase | Down | 0.76 | YM6 | 56.02 |
| A0A1D5XQP7 | Protein disulfide-isomerase | Down | 0.83 | YM6 | 54.19 |

## DISCUSSION

It has been demonstrated that seed deterioration is accompanied by accumulation of reactive oxygen species (ROS), lipid peroxidation, inefficient energy supply and degradation of stored proteins and nucleic acids (*Lee et al., 2012*; *McDonald, 1999*; *Rajjou et al., 2008*; *Yao et al., 2012*). Our present work showed that with extended storage time, increased number of DEPs were mainly involved in nutrient reservoir activity, enzyme activity and regulation, defense/stress response, and other processes in aged seeds (65 in YM45, 74 in YM28, and 871 in YM6) compared with untreated seeds (YM97), which resulted in gradually decreased germination rates. Previous reports also indicated that physiological properties in wheat can vary with the degree of hardness during accelerated ageing. Additionally, the post-harvest storage-tolerance of soft and hard wheat was different under the same storage conditions (*Keskin, Yalçın & Özkaya, 2018*; *Zhang et al., 2017*). Our current results showed that the Gr of soft wheat seeds of the 'Yangmai 15' cultivar was higher than that of hard wheat seeds of the 'Aikang 58' cultivar under the same storage conditions (*Lv et al., 2016*), confirming the superior post-harvest storage-tolerance of soft wheat. Accompanying analysis of the physiological changes, we performed a TMT-based dynamic quantitative analysis of proteomic changes in the 'Yangmai 15' soft wheat cultivar during artificial ageing.

Wheat grain proteins accumulate during seed development and break down during seed ageing. Our results indicated that the reduced abundance of storage proteins associated with processing qualities (*Rasheed et al., 2014*) was consistent with that in hard wheat during artificial ageing (*Lv et al., 2016*). The decreased abundance of storage proteins might lead to an inefficient energy supply for seed germination. Additionally, oleosins that were previously found to be downregulated in hard wheat and are known to function in germination (*Huang, 1992*) were also downregulated in the present work, consistent with the reduced Gr. A previous study showed that alpha-amylase/trypsin inhibitors are expressed during germination and following wounding, and they are beneficial to organ development (*Dong et al., 2015*). In our current results, the reduced abundance

of seven alpha-amylase/trypsin inhibitors and a subtilisin-chymotrypsin serine protease inhibitor together with the increased activities of amylases and proteases during artificial ageing might indicate that protection of wheat seeds is gradually diminished during seed deterioration. Additionally, the upregulated acyl-coenzyme A oxidase, responsible for β-oxidation (*Goepfert & Poirier, 2007*), was also upregulated during ageing, further indicating seed deterioration.

During accelerated ageing, proteins involved in electron transport, signal transduction, and energy and metabolism were differentially regulated. In animal cells, overexpression of cytochrome c oxidase, which functions in the final step of the electron transport chain, could lead to cell apoptosis (*Sanchez-Alcazar et al., 2000*). In the present work, cytochrome c oxidase subunit was upregulated, as was observed previously in maize seeds during accelerated ageing (*Xin et al., 2011*). Protein phosphorylation plays a key role in many signaling pathways including the responses to stresses such as heat shock, desiccation, peroxide and high salt stress, and calcium-dependent protein kinases and 14-3-3 proteins could facilitate the phosphorylation of target proteins (*Trewavas, 2000*). Their upregulation in the present study suggested that signal transduction may be disturbed, as occurs in maize seeds following accelerated ageing (*Xin et al., 2011*). In the present work, glyceraldehyde-3-phosphate dehydrogenase, malate dehydrogenase and succinate dehydrogenase that are involved in glycolysis (EMP) and the tricarboxylic acid cycle (TCA) were downregulated, along with two ATP synthase subunits in accordance with changes known to occur in aged maize and hard wheat (*Lv et al., 2016*; *Xin et al., 2011*). Compared to the downregulated enzymes associated glycolytic pathway-tricarboxylic acid cycle in hard wheat (*Lv et al., 2016*), several enzymes involved in glycolysis, the pentose phosphate pathway, and the tricarboxylic acid cycle were upregulated, including citrate synthase, isocitrate dehydrogenase, glucose-6-phosphate isomerase, 6-phosphofructokinase, and pyruvate kinase (Table 1). The EMP-TCA provides energy and carbon skeletons for other metabolic processes during seed imbibition, and a recent report demonstrated that storage-tolerant seeds might have higher EMP-TCA activity than storage-sensitive seeds (*Chen et al., 2018*). These results indicated that soft wheat seeds might have higher EMP-TCA activity than hard wheat seeds, indicating superior post-harvest storage-tolerance for soft wheat seeds.

To protect grains against various stresses such as oxidative stress, high temperature, and pathogen infection, several stress- and defense-related proteins are regulated during artificial ageing. Previous studies indicated that oxidative damage occurs during seed storage (*Galleschi et al., 2002*; *Calucci et al., 2004*). Seeds possess defense systems such as such as ascorbate and glutathione cycle to scavenge ROS and prevent oxidative damage. Thus, the storability of seeds is believed to be associated with the capacity to detoxify ROS. A decreased Gr is associated with an increase in ROS and a decrease in the activities of antioxidant enzymes such as peroxidases, catalases, ascorbate peroxidase, glutathione reductase, superoxide dismutase and glutathione-related enzymes (*Goel, Goel & Sheoran, 2003*; *Xin et al., 2014*). In our current work, superoxide dismutase, L-ascorbate peroxidase and three peroxidases were downregulated, consistent with reports in aged maize, soybean seeds, and hard wheat seeds (*Lv et al., 2016*; *Xin et al., 2011*; *Xin et al., 2014*). However, our

present study also indicated that glutathione S-transferase (GST) and glutathione synthetase were upregulated compared with levels in hard wheat (*Lv et al., 2016*), and these proteins were assigned to ROS detoxification and stress response function. The results indicated that soft wheat seeds might possess a stronger antioxidant defense system than hard wheat seeds, consistent with a recent report showing that storage-tolerant seeds utilize GSTs for ROS detoxification and oxidative stress tolerance (*Chen et al., 2018*). These results might also help to explain why soft wheat displays superior post-harvest storage-tolerance storage compared with hard cultivars under the same storage conditions. Additionally, upregulation of proteins such as pathogenesis-related proteins and wheatwin2 in soft wheat served to protect against fungal invasion (*Ng, 2004*) during accelerated ageing might also contribute to the superior post-harvest storage-tolerance of soft wheat. DNA and histone proteins are important components of chromosome, and previous reports revealed accumulation of chromosome damage under different storage conditions (*Rao, Roberts & Ellis, 1987*; *McDonald, 1999*). In our present work, decreased histone H2A/B levels during ageing might be responsible for chromosomal damage (*Rao, Roberts & Ellis, 1987*), and decreased abundance of 40/60S ribosomal proteins and unfolded protein response-associated protein disulfide-isomerases indicated diminished protein biosynthesis capacity (*Kimura et al., 2015*).

To investigate the mechanism of seed ageing, accelerated ageing conditions in which seeds deteriorate under high temperature and relative humidity were extensively employed to simulate natural ageing. Some authors questioned the physiological relevance of artificial ageing for short durations, whereas others concluded that controlled deterioration treatment can mimic molecular and biochemical events that occur during natural seed ageing (*Galleschi et al., 2002*; *McDonald, 1999*; *Rajjou et al., 2008*). Although accelerated ageing is unable to fully mimic natural ageing, our present work provides dynamic proteomic insight into soft wheat seeds during accelerated ageing.

## CONCLUSIONS

In this study, dynamic physiological and proteomic changes in soft wheat seeds that occur during accelerated ageing were investigated. The upregulated defense/stress-related proteins including glutathione S-transferase, glutathione synthetase, wheatwin-2 and pathogenesis-related proteins identified herein might be responsible for the superior post-harvest storage-tolerance compared with hard cultivars under the same storage conditions. Our results identified potential target proteins underlying the physiological changes and improved post-harvest storage-tolerance of soft wheat during seed deterioration.

### Funding

This work was supported by grants from the Natural Science Foundation of China (31371850, 31772023 and 31501575), the Natural Science Foundation of Henan province (162300410047), and the Key Program of Science and Technology Development of Henan

province (162102210191). The funders had no role in study design, data collection and analysis, decision to publish, or preparation of the manuscript.

### Grant Disclosures

The following grant information was disclosed by the authors:

Natural Science Foundation of China: 31371850, 31772023, 31501575.

Natural Science Foundation of Henan province: 162300410047.

Key Program of Science and Technology Development of Henan province: 162102210191.

### Competing Interests

The authors declare there are no competing interests.

### Author Contributions

- Yangyong Lv performed the experiments, prepared figures and/or tables, authored or reviewed drafts of the paper.
- Pingping Tian and Shuaibing Zhang analyzed the data.
- Jinshui Wang contributed reagents/materials/analysis tools.
- Yuansen Hu conceived and designed the experiments, approved the final draft.

### Data Availability

The raw data and the entire annotated MS/MS spectrum are available at the ProteomeXchange Consortium via the PRIDE partner repository with the dataset identifier PXD009156.

### Supplemental Information

Supplemental information for this article can be found online at http://dx.doi.org/10.7717/peerj.5874#supplemental-information.

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
