# Peer review of "Dynamic proteomic changes in soft wheat seeds during accelerated ageing"

_PeerJ, doi:10.7717/peerj.5874_

## Round 0.1 · original submission · Major Revisions

Experts in the field have carefully reviewed your manuscript and, although they found merit in your study, they have raised a number of concerns that preclude acceptance in its present form.

I invite you to respond to the reviewers' comments, included at the bottom of this letter, and to revise your paper accordingly. The revised version will be re-evaluated by the original reviewers.

Reviewer 1 ·

Basic reporting

This paper reports on the proteomics changes during accelerated aging or different grain germination rates. It is an important topic and an interesting article. However, I raised some points that need to be clarified by the authors. This paper has a high similarity with Lv et al., 2016 by the same authors and the originality of the current paper need to be also explained. Also, a discussion between the differential expressed proteins and their respective discussion need to be included, because to better comprehend the effect of different germination rates in the proteomic was the main objective.
The term “storage tolerance” can be very sound for the authors but talking about wheat research, the term storage can mean different things, and I suggest the authors adopt another term, such as “post-harvest storage-tolerance or wheat seed storage tolerance”.
Abstract
Line 20-22. This comparison appears incomplete “more than” what? Please re-write.

Introduction
Line 29-30: Why the wheat seeds storage in China is carried out in horizontal warehouses and why for so long? I think that a simple sentence here could help the reader to better understand the justification and practical application of the work, since in the other countries the way how seeds are storage is completely different.
Line 44-45: In this sentence, it means that germination ratio and physiological enzymatic activities varied with hardness when seeds were submitted to long-term storage or during grain storage?
Line 51-52: The authors cited some previous works of the group, and they justified this current work saying that “the dynamic physiological and quantitative proteomic changes in soft wheat seeds that take place during accelerated aging remain unclear and, need to be elucidated”, but it is not clear in which aspect the previous works needing better comprehension, not only for the authors group, but taking account the literature. For example, in 2016 Lv et al. showed 162 differentially expressed proteins during artificial ageing. Please be clear about the originality of this work.

Experimental design

Line 59: Please provide more information about the classification of “soft cultivars”. Is it Triticum aestivum? Yangmai 15’ represents only one cultivar or variety? Is it a soft common/bread wheat, like soft US wheats?
Line 60: Please explain what “original germination rate” is or how it can be achieved.
Line 61: I think this is the average of seed moisture content.
Line 66: What do the authors want to say with: “seed samples with three biological replicates”? It means three seeds, three lots, please make it clear.
Line 75: Since the main objective of this work is to perform a tandem mass tag-based quantitative proteomic analysis, is essential that the authors explain this methodological step, because it is not clear. For example: the samples were labeled and put all together ?
Moreover the sentence: “Peptides from YM97, YM45, 76 YM28, and YM6 were labelled with 126, 127, 130, and 131, respectively” is not clear. Please specify.

Line 87: Since the MS method used is based on DDA acquisition, please provide methodological details of how the peptides were selected for MS/MS.
SDS-PAGE analysis is presented in the results but not mentioned on the methods. Please include.

Validity of the findings

Results
Line 113: About SDS-PAGE analysis, there is no comment in the results? Why there is a difference of intensity between samples if for all sample 30 ug were loaded?
For LC-MS/MS analysis the sample were injected in the same concentration?
Line 120: What criteria were used to quantify 4509/6271 proteins?
From the quantified proteins and DEP how many are uncharacterized?
The results section is basically descriptive, I think it will be useful to highlight differences between samples, it means between germination rates, since in the discussion section the authors did rather a comparison with the literature, and an interesting discussion about the DEP but not explain the difference of the DEP in different germination rates that remains unclear.
For example, in the conclusion it is stated “1010 DEPs were identified, most of which were present when Gr was 6%.” This statement is not clear presented in the results and discussion.

Lines 146-149: For example, in the Figure 2 C the DEP belonging to molecular function of YM97 was extremely different from the others. How can the authors explain this result?
In the same way, in lines 150-153 there are differences between GR28% and GR6%, what is the practical comprehension? How these results can be interpreted concerning storage tolerance? The alpha-amylase is very important for the wheat quality and it is expected that there is an increase when germination occurs.
Line 169-172: This result was found for all samples? Why is inconsistent? The enzymes were up-regulated and their inhibitors were down expressed…
Discussion
Line 194-196: Again there is an incomplete comparison ‘Yangmai 15’ was higher than that of ‘Aikang 58 in which terms? Why these results confirmed the superior tolerance of soft cultivars?

Line 242-243: Why the results involving Glutathione related enzymes were unexpected?

Lines 246-248: It is not clear which traits/features lead to superior storage tolerance.

Figures: Please give a better explanation in the legend about the comparison between the differentially expressed proteins: for instance what does it means YM6/97 in up-regulated, means that proteins are up regulated in Gr6% compared to Gr97%. It I not clear in the legend neither in the text.

Additional comments

No comment.

Reviewer 2 ·

Basic reporting

Wheat is one of the major food crops of the world. However, little is known about proteome of different wheat cultivars during different conditions. Here, Lv et al performed a detailed analysis of the proteome of aging wheat of soft wheat cultivar. The quantitative proteomics analysis revealed some significant differences with respect to protein abundance. In addition, the biochemical analysis reveals differences in respiration and germination of the aging soft wheat. This paves the way for further in depth, quantitative (phospho)proteome-wide differential analyses upon aging/stress/pathogen attack or environmental change. The manuscript introduction is clear. Literature is well referenced. Figures are relevant, well-labelled and described.

Experimental design

The authors have done commendable job in defining the research question. It is clear the work by Lv et al fills an important knowledge gap.

Validity of the findings

The methods are sound and details have been provided. Overall investigation performed is technically sound. Conclusion are well stated and limited to supporting results. The major problem is authors only look upto 180 days storage. Wheat is routinely stored for years and it is important to study aging in wheat stored for 1-2 years. The authors should mention limitation of the study in abstract and discussion.

Additional comments

Specific comments:
1. The authors use inconsistently either ageing/aging. Please use one word consistently throughout the manuscript.
2. In Discussion Line194 authors suggest “Our current results….” While comparing Yangmai 15 and Aikang 58. These strains are not compared here.
3. There are problems with references formatting “Lv…Plos One,” and “Pasha….Int.”
4. In Table 1 the authors should provide values (ratio) of upregulated and downregulated proteins rather than just mentioning regulated type

Reviewer 3 ·

Basic reporting

The research is innovative and comprehensively described the proteomic changes in soft wheat seeds during storage through accelerated / artificial aging. The research will give basics for further investigation of physiological and biochemical changes in potential target proteins.
The results have been described in clear and unambiguous language that is understandable by the readers.
Hypothesis is very clearly explained

Experimental design

The primary research objectives have been clearly addressed in material and methods. The methods have been well designed and justified

Validity of the findings

The research work have some novel findings and the data is pretty useful for others. The manuscript has been well concluded.

Additional comments

1. Introduction and review have been described in clear and unambiguous language that is understandable by the readers. However, there are chances for improvement at some places like in line number 44-45 (Zhang et al., 2017), the sentence should be re-written for clear perception.
2. As described in material and methods that authors have conducted first tandem mass tag (TMT) -based dynamic proteomic analysis in lines 53-54, but there is no detail for importance of using this technique in the manuscript. Provide more information regarding the said technique.
3. In lines 128-131, no reference method is provided for protein annotation, GO, Domain and pathway analysis.
4. Under the heading of discussion, the reference used in line number 212 (Aoyama et al., 1994) is old one and also not related to seed storage, research was related to rat hepatoma cell and immunoblot analysis conducted to study the peroxysomal fatty acid β oxidation enzyme. Reference should be reviewed.
5. In discussion part there is contradiction b/w two statements regarding "up regulated" and "down regulated " enzymes in tricarboxylic acid cycle. line 226-229.
6. There is ambiguity in word "themselves" in line 231 under discussion. It should be replaced with suitable noun like soft wheat seeds or grains etc. according to the manuscript.
7. The reference mentioned in line 244 (Wang et al., 2015) is not related to research commodity because it is related to perishable commodity i.e., tomato that is not seems suitable for this work. Ref should be replaced with suitable reference related to seed storage study.
8. Reference is missing In lines 248-251 where the behavior of histone H2A/B has been described. Add some suitable reference regarding this change that occur during aging.
9. Overall the work is comprehensively described for proteomic changes in seed of soft wheat variety which will help in further research in deteriorative changes in storage grains.

---

## Round 0.2 · accepted · Accept

The revised manuscript answered well all referees´ comments and is now acceptable for publication, with the minor English edit suggested by reviewer 2 to be corrected during proofs.

Reviewer 2 ·

Basic reporting

Self-contained with relevant results to hypotheses.

Experimental design

Research question well defined, relevant & meaningful. It is stated how research fills an identified knowledge gap.

Validity of the findings

Conclusion are well stated, linked to original research question & limited to supporting results.

Additional comments

Wheat is one of the major food crops of the world. However, little is known about proteome of different wheat cultivars during different conditions. Here, Lv et al performed a detailed analysis of the proteome of aging wheat of soft wheat cultivar. The quantitative proteomics analysis revealed some significant differences with respect to protein abundance. In addition, the biochemical analysis reveals differences in respiration and germination of the aging soft wheat. This paves the way for further in depth, quantitative (phospho)proteome-wide differential analyses upon stress/pathogen attack or environmental change. I have some minor comments.

1. The title is somewhat vague and does not justify the study. Since in this study the authors have compared just two wheat strains and a more rigorous analysis is needed to verify these results with other wheat varieties. I believe limitation of the study should be mentioned.


2. Improve English usage in manuscript. I will cite few examples
a. Line 140 “indicating that the protein was not degraded and hence” It should be “indicating that the proteins were not degraded and hence”
b. Abstract : “…..response to stimulus functions, consistent with processes occurring in hard”